



# Analysis on Multi-Factor Synergistic Hazards Mechanism of Wet Micro-downburst: Multi-Source Data Fusion Analysis based on Passenger Ship Capsizing Accident in Qianxi Region, Guizhou on May 4th,2025

Ankun Wu[1], Xi Guo[1], Dan Yao[2], Xiaoling Du[3], Dongpo He[3], Jue Li[1], Shisheng Jin[1], Tienan Bai[1]

[1]GuiZhou Meteorological Administration Data Center, Guiyang 550002, China

[2]Meteorological Observation Center of CMA, Beijing 100081, China

[3]Meteorological Observatory of Guizhou, Guiyang 550002, China

*Correspondence to*: Ankun Wu (wak-mail@163.com)

**Abstract.**To reveal the multi-factor synergistic hazards mechanism of downbursts, this study takes the passenger ship capsizing accident at Dongfeng Reservoir in Qianxi, Guizhou Province around 08:32 (UTC) on May 4, 2025 ("May 4" accident) as the research object, and utilizes the data of integrates radar, ground observation, lightning monitoring, video and on-site disaster trace data to systematically analyze the hazard process of wet microburst. A strong persistence intense radar echoes (≥60dBZ) was observed over the incident area. At 08:29 (UTC), the core intense echoes broke through the zero-degree layer and rapidly descended from 6–8km to 2–4km, which synchronized with the strongest downdraft, accompanied by localized high differential reflectance at 2–4 km showed a local high value $R > 2dB$) . After the intense echoes touched the ground, an explosive near-surface divergent flow field was triggered: wind speed surged from 0.8m/s to 34.7m/s within 6 minutes, temperature dropped by 14.9°C in 10 minutes, and air pressure jumped by 5.4hPa in 5 minutes, and cumulative rainfall reached 40.6mm. Lightning activity was dominated by cloud flashes (85.1%), with high-density areas coincided with convective paths. The phase transition of ice particles (indicated by ZDR) intensified downdrafts through "mass loading - cooling enhancement effect" feedback, which triggered wind direction reversal and the occurrence of hailfall. The distribution of disaster traces (directional lodging, quadrant-switching damage) spatial-temporal matched with the wind field evolution, which verifying the coupling disaster-causing mechanism of dynamic attenuation and microphysical transport.

## 1 Introduction

Downburst, as a highly destructive phenomenon in severe convective weather, it has a complex formation mechanisms and tremendous hazards. Fujita (1976) first proposed the concept in 1976 to describe the intense downward airflow and outward explosive divergent flow of the storm that caused the aircraft crash at New York's John F. Kennedy International Airport in 1975, since that, downburst has attracted extensive attention and in-depth research in the meteorological community (Fujita and Byers, 1977). Downburst refers to an explosive divergent outflow at the ground or near the ground triggered by strong downdrafts from convection cells with destructive power reaching F3 intensity (70–92m/s), which has comparable damage degree as tornadoes (Fujita, 1981).

In the field of meteorology, downbursts are classified into microbursts (4m–4km) and microbursts (4–40km) by horizontal scales. At the same time, downbursts are further categorized into dry microbursts and wet microbursts by precipitation conditions during strong winds (Fujita, 1977). In the aviation field, downbursts specifically refer to localized intense downward divergent airflow at 100m above ground level which exceeds the takeoff and landing speed (3–4m/s) , and these are predominantly small-scale micro-downburst (Xu et al., 2001).



The suddenness and local strong wind characteristics of downbursts pose a significant threat to transportation safety. Apart from the aviation field, ships on open waters are more vulnerable to the influence of strong wind divergent flow due to the

lack of terrain obstruction, which may, lead to loss of course control or even capsize.

Incidents such as the "Tragic Sinking of the Eastern Star" in Jianli, the Ningbo leisure fishing boat capsizing accident, and the Liuzhi Zangke River passenger ship capsizing accident, etc, water disasters Meng et al., 2016; Gu et al., 2025). have been closely related to strong convective winds wet micro-downbursts through field observations and disaster damage investigations (Wang et al., 2023). The coupled effect of intense precipitation and strong winds not only reduces visibility but

also exacerbates ship maneuvering difficulties by altering water surface friction. However, the analysis of downburst currents in existing studies primarily relies on macroscopic means such as weather radar, satellite remote sensing, and background field weather patterns analyses. Owing to its strong locality, small impact range, and rapid development, it often lacks micro-scale data such as high spatiotemporal resolution observations of surface meteorological elements, on-site disaster videos, which hinder the precisely depict its triggering, development, and disaster-response processes. Post-incident

analysis of the "Tragic Sinking of the Eastern Star" incident has confirmed that traditional single-station observations are difficult to capture the rapid evolution characteristics of wet micro-downburst, and multi-source data fusion technology is a critical solution to this limitation (Zheng et al., 2016).

On May 4, 2025, a major passenger ship capsizing accident occurred in the Dongfeng Reservoir of Qianxi, Guizhou Province, resulting in the capsized of four passenger ships ("May 4" accident). Preliminary investigations indicated the

accident has a close association with a wet micro-downburst. This incident is similar to the capsize of "Tragic Sinking of the Eastern Star" in 2015 due to a downburst, which re-emphasizes the fatal threat of downbursts to maritime navigation safety.

Compared to dry microbursts, wet micro-downbursts exhibit radar echo intensity exceeding 35 dBZ or the surface precipitation is greater than 0.25 mm, and the presence of hail or graupel particles aloft. Intense downdrafts arise from negative buoyancy generated by complex hydrometeor phase changes, including raindrop evaporation, hail melting, and

evaporation after hail melting (Fu et al., 2007; Proctor et al., 1989; Wakimoto et al., 1988). To comprehensively reveal the disaster-causing mechanism of the wet micro-downburst in the "May,4 accident", this study integrates multi-source data—including weather radar, automatic weather stations, lightning monitoring, video surveillance, and on-site disaster traces to deeply analyze the formation, development, and disaster-causing process of the wet micro-downburst. It aims to clarify its evolutionary law and disaster-causing mechanisms, providing scientific support for the prevention and response to

similar disasters, so as to enhance and guarantee the safety of water traffic capabilities, and reduce the losses from extreme weather events.

## 2 Data Description

The data utilized in this article mainly covering multi-dimensional information such as meteorological observations and actual disaster conditions.

1.Weather radar data. It includes multi-band weather radar composite reflectivity factor provided by the Guizhou Provincial Meteorological Data Center, as well as reflectivity factor (REF), radial velocity (VEL), and differential reflectivity (ZDR) data from the Guiyang S-band weather radar, which are used to analyze the echo characteristics and wind field structure of downburst.

2.Surface meteorological observation Data. Minute-by-minute data (temperature, humidity, pressure, precipitation, and

wind speed/direction) from automatic meteorological stations surrounding the incident area. Since the operational sampling frequency for some elements is 5 minutes, higher temporal resolution records were obtained through remote sampling from device-side storage. Data within 3 hours before and after the event were selected, with a focus on extracting 1-minute resolution of wind speed/direction, air temperature, pressure, and rainfall data from the "Tianqing" Big Data Meteorological Cloud Platform, which aims to capture abrupt changes in surface elements during the impact of downburst.



3. Other supplementary data. It includes lightning monitoring data from the Guizhou Province 3D Lightning Monitoring Network, which is used to correlate the spatiotemporal characteristics of severe convective activities. Additionally, video surveillance data along the path of weather process and on-site disaster damage investigation data were collected, providing direct evidence for analyzing the disaster-causing process and impact range of the downburst.

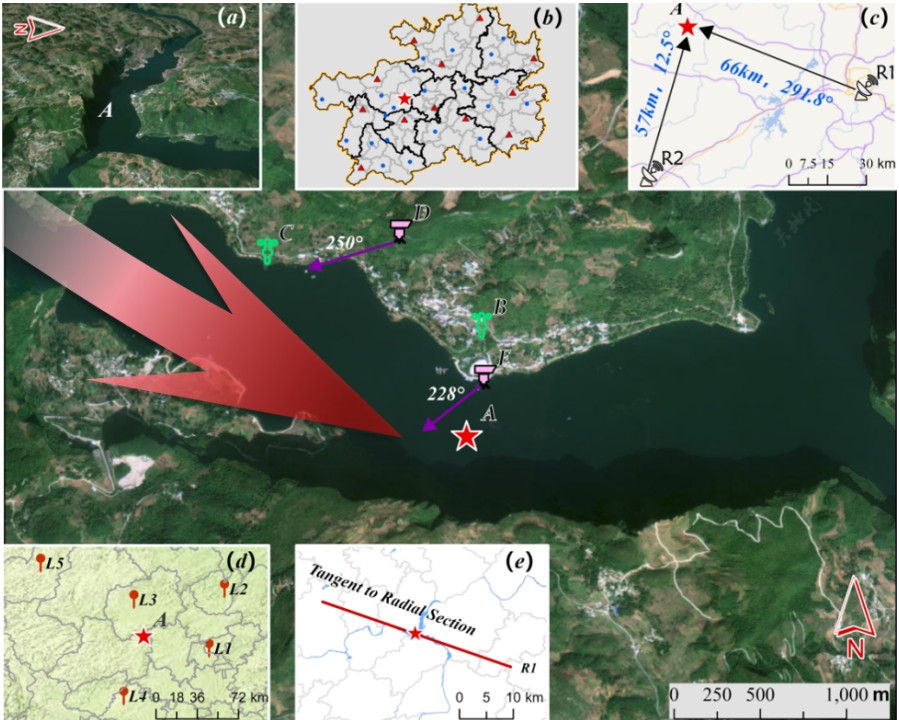

**Figure 1: Distribution of Meteorological Observations and Video Surveillance in the Incident Area and Surroundings.**

**A is water area where the passenger ship capsized. B and C are nearby automatic weather stations R7676 and R7657, respectively. D is Shoubayan surveillance camera, located at Wangjiang Platform on the Migratory Ancient Path, with an altitude of approximately 1263m and monitoring direction 250°. E is wharf square camera, which at an altitude of approximately 940m, with a monitoring direction 228°. Figure (a). The incident water area is a confluence of three rivers. Vertical limestone cliffs (200**

**－300 m high) stand like screens to the south and east, and the north side lies a gentle-slope forest and a terraced artificial wharf square carved out in accordance with the mountain terrain. The water surface altitude is approximately 912m. Figure (b). Layout of 12 S/C-band and 21 X-band weather radars in Guizhou. Figure (c). Locations, distances, and orientations of 2 S-band dual-polarization weather radars near the incident area. Figure (d). L1-L5 lightning monitoring network coverage around the incident area. Figure (e). Radial cross-section tangent of the Guiyang radar along the incident area.(The base map image is**

**derived from Tianditu images, with the map review number: GS (2024)0568.)**

## 3 Weather Conditions

From 06:00 to 12:00 on May 4th, scattered severe convective weather occurred along the northwest to southeast line of Guizhou Province, mainly characterized by thunderstorm gales, short-term heavy precipitation, and locally accompanied by hail process. A total of 31 stations across the province experienced thunderstorm gales of force level 8 (17.2 m/s) or above,

mainly concentrated in the east-central part of Bijie City, western part of Zunyi City, Guiyang City, and central part of Qiannan Prefecture. In Qianxi City of Bijie, two stations had wind speeds of force level 10 (24.5 m/s) or above. Among them, the Huawu Village Meteorological Observatory (Point C in Figure 1) recorded an instantaneous wind gale of 44.7 m/s (exceeding wind force level 14) between 08:32 and 08:33. Hail was observed at 40 stations, with the maximum hail diameter of 60mm in Majiang County, the southeast of province. Short-term heavy precipitation occurred at 38 stations, with the

highest precipitation intensity of 52.9mm/h recorded at the Chanong Station in Duyun City. The maximum cumulative



precipitation within 6 hours was 53.2 mm, which is observed in Gulong Town of Longli County.

This weather process entered through the northwest entrance of the incident water area at approximately 08:24, and affected about 25 minutes before moving out from the southeast entrance. During this period, extreme winds accompanied this weather process, which was the highest wind speed observed in Guizhou since 1961 when we had complete meteorological records.

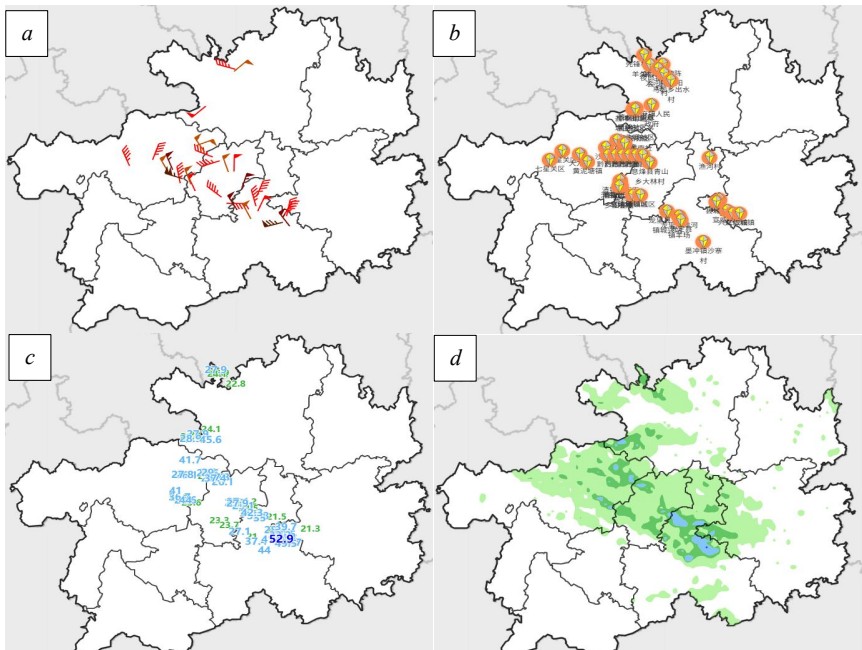

**Figure 2: Real-time Weather Conditions. (a) Thunderstorm Gales of Force 8 and Above.(b) Hail.(c) Short-duration Heavy Precipitation.(d) Accumulated Precipitation**

## 4 Results Analysis

### 4.1 Evolution Characteristics of Weather Radar Detection

The analysis based on composite reflectivity mosaics from 33 radars (Figure 1.b) at different times between 08:18 and 08:48 on May 4, 2025 indicated that the edge of strong echoes approached the incident area and the range of strong echoes (>65 dBZ) at initial stage was limited, which concentrated in the Guanyindong-Yangchang junction zone (northwest of the incident location). Convective cells were in the initial development stage which laid the foundation for subsequent strong echo system formation. From 08:18 to 08:24, strong echoes moved southeastward in cirrocumulus pattern with the reflectivity field expanding and intensifying. Dual high-reflectivity cores formed in the Yangchang-Xinren junction area, which indicated the convective system had entered a stage of rapid development and organization phase with significantly increased echo energy density.

From 08:24 to 08:30, the main body of strong echoes covered the incident area with the echo pattern merging from dual-core structure into a "crescent shape" accompanied by secondary echo center weaker than 65 dBZ. This reflected the convective system readjustments in dynamic and microphysical processes, which resulting in stable coverage of strong echoes over the incident area. From 08:30 to 08:42, high-reflectivity cores continuously acted on the incident area with the intensity of the strong reflectivity field (up to 74dBZ) and spatial range remain stable. The convective system entered into an efficient and continuous stage, providing a critical dynamic environment for downburst disaster formation. From 08:42 to



08:48, the main strong echo subject gradually moved away from the incident area, and the influence of high-reflectivity cores was lifed.

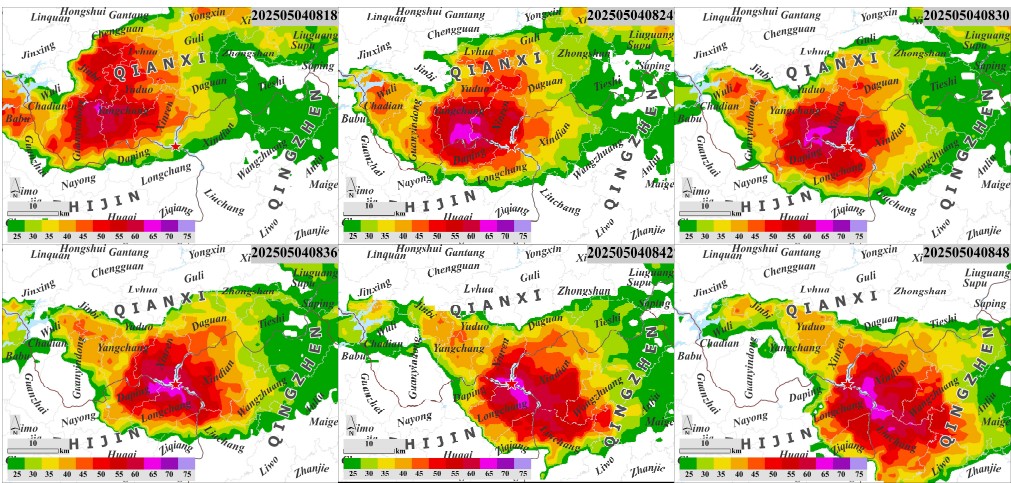

**Figure 3: Evolution of Weather Radar Echoes**

Due to the northwest-southeast movement path of convective cells, it is aligned with the radial direction of the Guiyang radar relative to the incident location, Figure 4 displays vertical profile of radar along the radial direction of the incident area (as shown in Figure 1.e) from 08:23 to 08:44, including composite reflectivity (REF, left column), differential reflectivity (ZDR, middle column), and radial velocity (VEL, right column). The red pentagram (A) marks the capsizing location of the passenger ship, the blue solid line (B) indicates the zero-degree layer height (approximately 8km) determined from surrounding sounding data, and the bottom curve (C) represents ground altitude changes along the radial direction. The evolution process of the wet microburst is analyzed from two aspects: microphysical characteristics and dynamic structure:

The left column REF profile shows that between 08:23 and 08:44, a strong echo core (intensity >50dBZ) persisted above the incident location (Point A), with its height gradually descending from an initial 6–8 km to 2–4 km. The vertical extension and rapid descent characteristics of strong echoes reflected the rapid growth and gravitational fall of hydrometeor particles (hail, graupel, etc.) within convective clouds, which directly related to the mass loading mechanism of downburst formation. The descent of abundant precipitation particles dragged the airflow, triggering and enhancing the descending motion (Proctor et al., 1989). The strong echo core reached its lowest height (near 2 km) between 08:29 and 08:34, which corresponding to the critical stage of explosive radial divergence of descending airflow hit the ground. This is highly consistent with the temporal characteristics of strong echoes steep descent and strong wind outbreaks on the ground observed by Zheng et al(2016). in the "Eastern Star" incident, revealing disaster-causing features commonalities of such processes.

The middle column ZDR profile reveals the phase evolution of precipitation particles. From 08:23 to 08:34, a high ZDR region (>2 dB, red area) appeared at 2–4 km altitude above Point A, which exactly located in the melting layer below the zero-degree layer (Line B, 5 km). High ZDR value typically indicates the dominance of flattened water droplets,combined with the zero-degree layer position, it can be inferred that upper-level ice-phase particles (hail, graupel) melted after passing through the zero-degree layer during their falling, forming abundant supercooled water droplets(Laroche,2002). After 08:34, the high value of ZDR region further descended to near the surface with the strong echo core, indicating that the melted water droplets continued to be transported to the ground, enhancing near-surface precipitation intensity and air cooling effects near the ground, and providing sustained momentum for downdraft airflow.

The right column VEL profile clearly depicts the dynamic structure of the downburst. After 08:23, a negative VEL region (toward the radar) appeared above Point A at 4–8 km (near the zero-degree layer height), indicating the dynamic power



activation of downdraft airflow in the convective storm middle layer. From 08:29 to 08:39, the negative VEL zone region in middle layer continued to extend downward, and the VEL at Point A shifted from negative to positive, indicating the impact

of downdraft airflow hit on the ground. The positive VEL zone region near-surface expanded and formed a divergent field with surrounding negative VEL, corresponding to the peak value of surface divergence of downburst with strong winds directly acted on the incident area. This process aligns with Fujita's (Fujita, 1985) classic model of "upper level outflow – middle level descend motion – surface divergence," confirming the strong wind disaster mechanism of the capsizing incident at Point A. After 08:44, the velocity of the descending airflow weakened with the divergence range narrowed, and the

dynamic structure gradually dissipated.

Furthermore, the persistent presence of the REF strong echo core above the the zero-degree layer (6–8 km) indicates sufficient growth of ice-phase particles (hail, graupel) at the same time, the development of high ZDR value regions below the zero-degree layer (2–4 km) confirms the dominant role of supercooled droplets formed by melting ice-phase particles. This chain process of "ice-phase growth – water-phase melting – evaporative cooling" provided sustained negative buoyancy

driving for descending airflow (Mahale et al., 2016). In particular, the strong echo core breaking through the zero-degree layer and rapidly descending at 08:29, which synchronized with the strongest descending airflow shown by VEL, verifying the enhancement effect of particle phase change on descending motion.

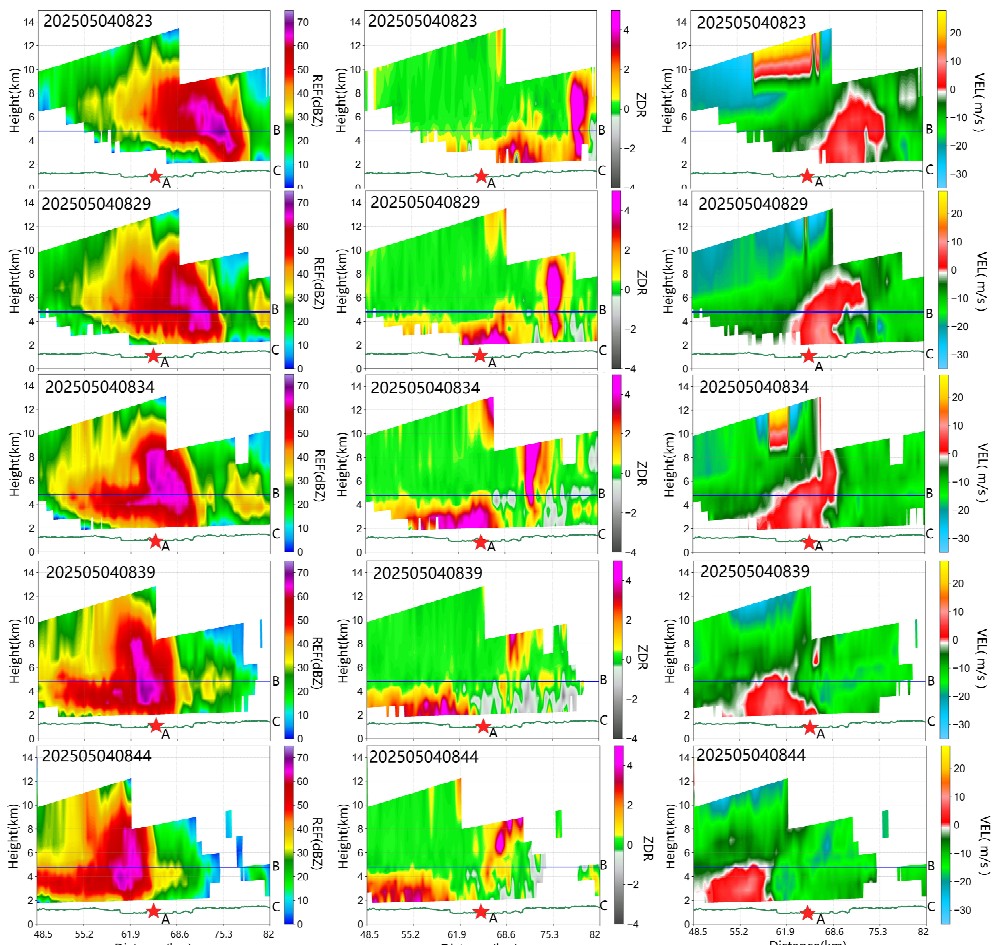

**Figure 4: Guiyang S-band Weather Radar Radial Vertical Profile. A indicates location of the passenger ship capsizing incident.B**

**indicates zero-degree line position determined from nearby 3-day sounding data.C indicates ground elevation changes along the**





**radial direction**

### 4.2 Evolution Characteristics of wind, temperature, and air pressure at Surface Automatic Stations

Before 08:23, the instantaneous wind speed at Station R7676 (indicated by the green polyline) fluctuated slightly within the range of 0–3 m/s, with wind direction (indicated by red arrows) stably pointed towards to the east direction. During this stage, warm and moist airflows were smoothly transported along the river surface, and the low-roughness underlying surface significantly reduced airflow disturbance, resulting in a weakly disturbed and highly stable wind field. From 08:23 to 08:26, a slight disturbance occurred in the wind field with wind speed dropping to the minimum value, and then followed by an explosive abrupt change. Starting from 0.8m/s at 08:26, wind speed continuously surged to the peak (exceeding 34.7m/s around 08:32), exhibiting typical "extremely strong wind" characteristics. Concurrently, wind direction sharply shifted from northeasterly (56°NE) to westerly (near 270°NW), with a crisply decisive and lag-free direction turning process. This combination of "wind speed jump + abrupt wind direction turning" is a classic indicator of cold outflows formed by thunderstorm cloud downdrafts impact on the ground (Orf et al., 1999).

From 08:32 to 08:38, wind speed gradually weakened to approximately 10m/s, maintaining a predominantly westerly direction. Subsequently, wind direction shifted to counterclockwise. It turned to southerly from 08:39 to 08:40, then easterly again after 08:41, with wind speed re-intensifying to 17.6m/s. After maintaining a wind speed of approximately10 m/s around 3 minutes, wind speed rapidly dropped below 5m/s at 08:46 as cold pool outflow energy dissipated, and wind direction returned to a weakly disturbed state. Station R7676 is adjacent to an open river surface with extremely low roughness of underlying surface, and the airflow is minimally disturbed by topographic obstructions. In contrast, surface station R7657 is restricted by geographical environment and ultrasonic wind measurement conditions, which recorded significantly lower wind speeds than R7676. however, both stations exhibited consistent wind direction trends.

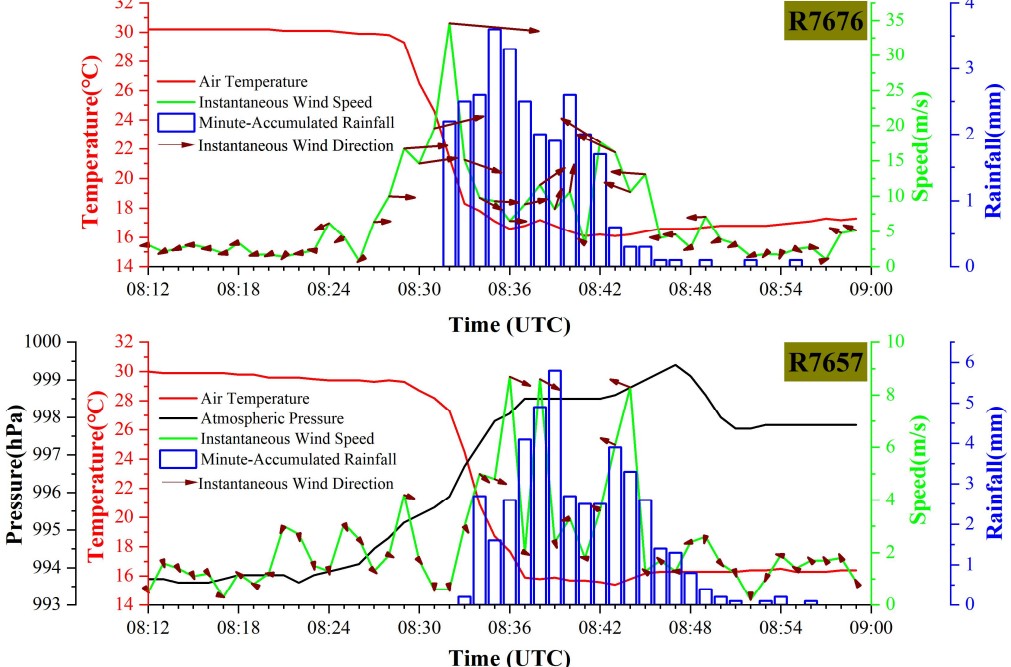

**Figure 5: Time Series of Meteorological Observation Elements at Huawucun Station (R7676) and Baili Gallery Station (R7657)**
Temperature changes (red curve) showed a clear temporal correlation with wind field abrupt changes. During the initial



stage of wind field abrupt changes (before 08:26), temperature remained stable at approximately 30°C. As cold pool outflow
continued to advance, temperature dropped sharply after 08:32, plummeting to 16.1°C within 10 minutes. The accompanying
small-scale cold pool was significant, with 10°C temperature difference between downburst and ambient environment
proposed by Atkins et al (1960). During this period, precipitation processes (blue bars represent minute cumulative rainfall)
occurred simultaneously. Cumulative precipitation at station R7676 reached 26.9mm within 10 minutes, while station R7657
recorded even higher cumulative precipitation of 40.6mm due to topographic uplifting and enhanced cold pool convergence,
confirming the intensifying effect of cold pool outflow on vertical transport of water vapor. Air pressure changes (black
curve) also exhibited typical high-pressure response characteristics of thunderstorms. Before 08:23, air pressure fluctuated at
a low level of 994hPa. As cold pool outflow hit the ground, high-density cold air rapidly accumulated near the surface,
which caused air pressure rose by 5.4hPa. This is consistent with observations by Mahale et al. (2016) and Liping Zhai et al.
(2019) that surface pressure increases associated with wet downbursts accompanied by heavy precipitation. The sudden
increase of air pressure, wind field abrupt changes, and temperature plummet were highly coincident in time dimention,
verifying the synergistic evolution mechanism of "dynamic disturbance–thermal response–air pressure adjustment" driven by
cold pool outflow, and providing multi-element observational evidence for the energy release process of severe convective
weather systems.

### 4.3 Lightning Activity Characteristics

Downburst is a common type of disastrous weather. Outfield experiments indicate that the occurrence of downburst can be
detected in 60%–80% of thunderstorms weather (Wakimoto, 2001). According to the data from lightning monitoring network,
1983 lightning flashes were detected near the incident area between 08:00 and 09:00, including 1688 cloud flashes
(accounting for over 85.1%). Lightning frequency ranged from 5 to 66 flashes/min with cloud-to-ground (CG) flashes
maintaining a low overall proportion (<25%). Most flashes occurred within clouds, attributed to "cold pools" formed by
225    downdrafts, which hinder the transimission and discharge of electric charges to the ground.

Spatially, lightning distributed in a northwest–southeast direction in a strip-like pattern, consistent with the movement
direction of downburst convective cells. Four high-density areas (>8 flashes/km²) formed along the strip in northwestern of
Jinbi, central of Yangchang, southern of Xinren, and northeastern of Longchang. Their occurrence times (approximately at
minutes 06, 20, 34, and 42 of 08:00) coincided with strong echo core centers, reflecting coupling between charge activity and
230    dynamic structure in the convective system. The incident site was located in a sub-high lightning density zone (green-yellow,
4–6 flashes/km²), which was in the "tail–transition zone" of the main lightning belt—the downdrafts outflow    region of
downburst. Although it is not the most lightning-dense area, it was directly affected by charge activity in the convective
system.



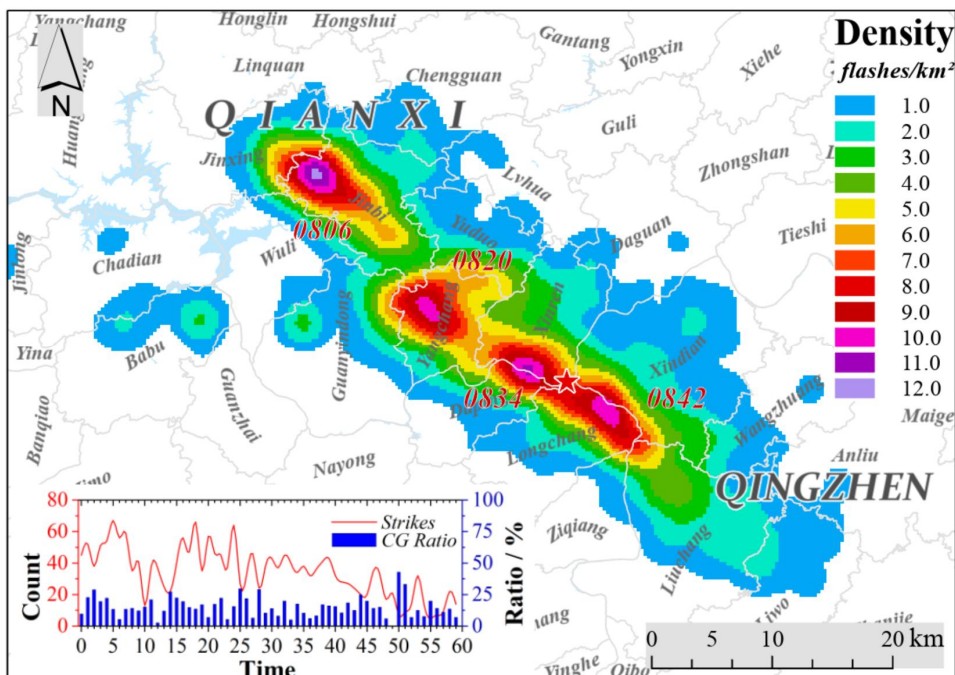

**Figure 6:Spatio-Temporal Distribution of Lightning Density (08:00-09:00)**

**4.4 Verification of Disaster-Causing Process via Video Surveillance and On-Site Disaster Traces**

**4.4.1 Temporal Correlation of Video Surveillance**

Based on real-time videos from surveillance cameras at Shoubayan (position D in Fig. 1) and the wharf (position F in Fig. 1), combined with collaborative analysis of meteorological elements. we observed that the main cumulonimbus body continuously moved into the monitoring field of view at 08:18. The high-layer cloud anvil exhibited cyclonic rotation characteristics (extending and spreading southeastward), and its bottom was accompanied by an explosive downward airflow forward—airflow propagation speed was significantly faster than that of cloud anvil advection. It indicated that downdraft momentum in the severe convection core area began to transmit downward, accumulating kinetic energy for downburst touchdown ground (Fig. 7a). At 08:24, the high-altitude cloud system underwent an abrupt morphological change (anvil structural distortion marked by red dashed lines), and mountain vegetation showed a weak westerly response. This corresponding to the touchdown of the upstream front of downburst downdrafts, where high-altitude momentum downward transfer airflow arrived before the main body, forming an initial wind field disturbance at the mountain top. At this time, downdrafts had not completely diverged yet, and the wind field was dominated by unidirectional advection, which is the initial signal "point-source divergence" of downburst (Fig. 7b). At 08:29, wharf cameras showed rapid advancement of rain and fog (water vapor phase transition triggered by downdraft descending cooling) with waves ≥1m were triggered on the river surface (red zone Z). The cruise ship flag and directions of wind and wave were dominated by northwest winds. It is verified that downburst frontal divergent airflow diverged from the convection center to southeastward, forming linear shock waves in the water area (Fig. 7e). At 08:32, the main body of strong wind reached Shoubayan (Fig. 7c), and the meteorological station R7676 recorded an instantaneous maximum wind speed of 44.7m/s.

Mountain-top vegetation significant tilted (with the northwest wind continuously dominating), and cold air carried by downdrafts mixed with ambient warm and moist air, triggering local rain and fog outbursts (visibility <10m). This stage corresponded to the energy peak of downburst divergent wind field, with the highest momentum transmission efficiency and



the greatest disaster risk. After 08:40, wind direction shifted from northwesterly to southeasterly dominance, which were visible on mountain top trees (Fig. 7d) and riverbank cruise ship flag (Fig. 7f), accompanied by the descent of small hailstones. This phenomenon "wind direction reversal + hail" resulted from the synergy of downburst cold pool divergence and ice-phase particle transport, where cold pool's low temperatures preserved upper-level incompletely melted graupel particles, which were transported to the surface by southeasterly winds. This process demonstrated the coupled disaster mechanism of dynamic downdrafts and microphysical processes.

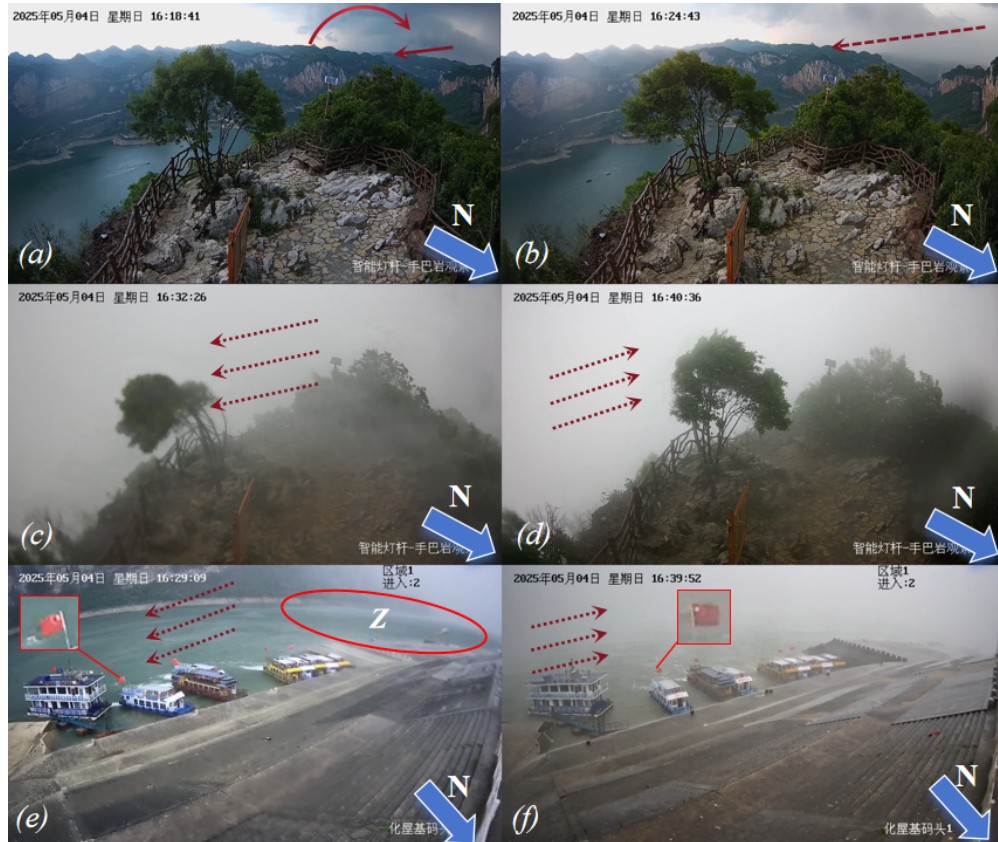

**Figure 7: On-Site Video Surveillance (The video is sourced from scenic area cameras and the Tianyan Security System. Researchers, as authorized members of the investigation team, have no involvement in copyright issues.)**

### 4.4.2 Spatial Response of On-Site Disaster Damage and Dynamic Wind Fields

Based on field disaster damage traversal and spatial correlation analysis, we found 5 significant damaged areas, which could verify the analysis. $Z_4$ is the first significant disaster area of the downburst upstream, trees showed "uprooting and directional breakage" with lodging/breakage directions eastly southeastly, which is highly consistent with the downburst movement path (northwest-southeast) and initial divergent wind field (northwesterly-dominated stage). Rapeseed "patchy lodging" reflected three-dimensional wind field disturbances (horizontal divergence + vertical disturbance force) after strong downdraft touchdown, verifying the initial damage characteristics "explosive outflow" of downburst. The scope and intensity of disaster damage in $Z_4$ (first touchdown zone) were significantly weaker than subsequent zones. Disaster damage $Z_2$ (riverbank group construction square) showed a "wind field quadrant switching" response. Guardrails and billboards exhibited "unidirectional collapse", and the direction of square trees lodging were northeast, corresponding to the "southeastern quadrant airflow" of the downburst divergent wind field. Cornfields showed "small hail damage," and trees



had "twisted but unbroken branches", confirming the residual effect of "weak dynamics force + ice-phase particles" in the outflow from the cold pool. The disaster damage in the $Z_3$ area is distributed in point-like pattern with non-directional lodging of trees and streetlights, reflecting multi-quadrant airflow confrontation of the strong divergent wind field. The disaster damage exhibited "omnidirectional divergence" characteristics. In the $Z_1$ area, trees showed patchy eastward lodging/breakage, consistent with the downburst movement path, demonstrating the directional momentum transfer of the marginal wind field. $Z_5$ (residential houses) west-east walls exhibited "wall peeling + hail impact marks", corresponding to the downburst" wind direction reversal process" — first affected by westerly winds (initial divergent northwest quadrant), then easterly winds (switched to southeast quadrant), with hail impacted walls successively with bidirectional wind fields.

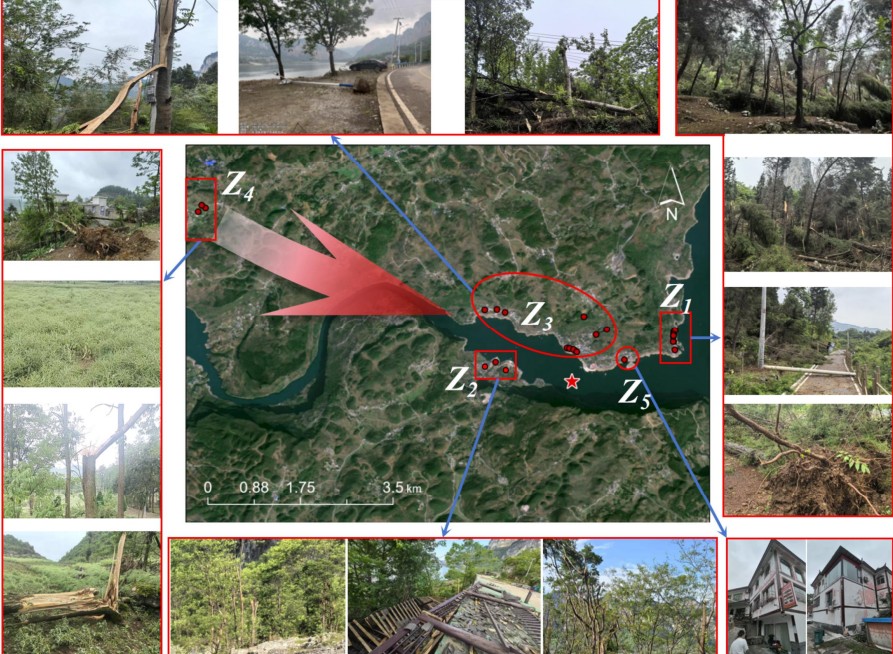

**Figure 8: Situation of On-Site Disaster Damage(The base map image is derived from Tianditu images, with the map review number: GS (2024)0568; the on-site photos are taken by the research and investigation team at the scene.)**

## 5. Conclusions and Discussion

Based on multi-source data including ground observations, lightning monitoring, video surveillance, and on-site disaster traces, this study systematically revealed the multi-factor synergistic evolution mechanism and disaster-causing process of the downburst event in the central of Guizhou on May 4, 2025. The main conclusions are as follows.

(1) Physical Mechanism of Radar-Ground Multi-Factor Synergistic Evolution

The intense echo core (>60 dBZ) rapidly descended from 6–8 km to 2–4 km (08:23–08:44), and reached its lowest altitude (near 2 km) between 08:29–08:34, which corresponded to the occurrence time of maximum ground extremely wind speed (34.7 m/s), confirming the temporal sequence law of "strong echo steep descent-ground wind disaster". A ZDR high-value zone (>2 dB) appeared at 2–4km below the zero-degree layer (5km), indicating dominance of flattened water droplets formed by melting ice particles, which enhanced the near-surface cooling effect. The downward extension of the negative VEL zone in the middle layer (4–8 km) triggered the expansion of the near-ground positive VEL zone, forming a dynamic structure of "middle layer descending initiation-ground divergence". Synchronous ground elements responses included: wind speed at Station R7676 jumping from 0.8m/s to 34.7m/s within 6 minutes, and the wind direction abruptly shifting from





northeast (56°NE) to northwest (270°NW). The cold pool outflow temperature caused a sudden drop of 14.9°C within 10 minutes and 5.4 hPa air pressure rose within 5 minutes due to thunderstorm high pressure. These three factors collectively constituted the "wind-temperature-pressure" coupling characteristics of downburst touchdown.

(2) Energy Coupling Mechanism of Lightning-Dynamics-Microphysics

During the downburst period (08:00–09:00), lightning activity presented characteristics of "high cloud flash proportion (85.1%) + strip-shaped distribution". The dominant cloud flash resulted from cold pool hindering charge transmission to the ground (cloud-to-ground CG flashes <25%). High lightning density areas (>8flashes/km²) were distributed along the northwest-southeast direction, which is spatially and temporally coinciding with convective cell paths and intense echo core

centers, verifying the coupling between charge activity and dynamic structure. The intense echo core descent of the REF (mass loading) and high-value ZDR zone (particle phase transition) jointly enhanced descent motion, triggering the "wind direction reversal + hail" phenomenon (after 08:40). Unmelted graupel particles fall to the ground with southeast winds, demonstrating the synergistic disaster-causing effect of cold pool and ice particle transport.

(3) Multi-Scale Verification of Disaster-Causing Process

Radar-video-ground collaborative verification showed that from 08:18–08:32, the vedio recorded cyclonic rotation of the cloud anvil, advancement of the descending airflow front (with a speed exceeding of the cloud anvil advection), and wind waves ≥1m on the river surface, which were highly synchronized with the time sequence of radar REF intense echo touchdown (from 08:29–08:34) and ground extreme wind speed(08:32). Spatial matching between disaster traces and wind fields revealed: "directional tree lodging" (east-southeast) in Zone $Z_4$ corresponding to initial divergent wind fields

(northwest wind dominance)."Wind direction quadrant switching" (northeast lodging) in Zone $Z_2$ consistent with VEL southeast quadrant airflow, and "bidirectional damage to west-east walls" of civilian houses in zone $Z_5$ directly reflected the wind direction reversal, verifying the coupling effect of cold pool divergence and dynamic wind fields.

The disaster-causing process of downbursts essentially involves complex coupling of dynamics, microphysics, and surface responses. In terms of dynamic mechanism, ground divergent wind fields wind direction changes and extreme wind speed

were dominantly triggered by middle-level downdraft airflow, and its "directional-omnidirectional" dynamic adjustment characteristics originated from quadrant switching of cold pool divergence, which reflecting the inherent logic of spatio-temporal wind field evolution. From the aspect of microphysical feedback, significant coupling exists between cloud flash-dominated charge activity and ice-phase particle transport. Spatial-temporal coincidence of high lightning density areas with convective paths confirms that charge activity regulated by dynamic structures, while mass loading and particle phase

transition enhanced descending motion, triggered "wind direction reversal + hail" phenomena, revealing that the cold pool's dual-path achieved enhancement of dynamic disaster-causing through "thermal protection-wind transport". In terms of surface response, the disaster damage distribution and morphology precisely map the evolution of wind field. Directional lodging, quadrant-switching collapse, and bidirectional building damage respectively correspond to initial divergent wind fields, quadrant airflow intervention, and wind direction reversal stages. Disaster differences between core and peripheral

zones quantitatively demonstrate dynamic attenuation characteristics, ultimately forming a closed-loop mechanism of "dynamic input-microphysical evolution-surface disaster damage output".

The study proposed multi-factor collaborative criteria, which break through limitations of traditional single-factor identification through coupling analysis of dynamics-microphysics-surface responses, providing new insights for severe convection warning in complex terrain areas. In the future, it is necessary to integrate high-resolution observations and

numerical simulations to deepen modulation mechanisms of terrain on cold pool paths and promote applications of multi-source data fusion methods in severe convection disaster mechanism studies.



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



**Author contributions**

WAK, YD, DXL, and HDP participated in the on-site accident investigation and completed the collection and organization of

relevant materials; WAK and GX were responsible for data analysis, chart plotting, and manuscript writing; LJ, JSS, and

BTN were responsible for the remote retrieval and processing of minute-level data from the observation equipment side.

390