# Peer review of "Analysis on Multi-Factor Synergistic Hazards Mechanism of Wet Micro-downburst: Multi-Source Data Fusion Analysis based on Passenger Ship Capsizing Accident in Qianxi Region, Guizhou on May 4th,2025"

_EGUsphere, 2025_

## Referee Comment (RC2)

**Review Comments**

Overall Assessment: The manuscript presents a detailed multi-source data fusion analysis of a wet micro-downburst event that caused a passenger ship capsizing in Guizhou, China. The topic is important for improving the understanding of localized severe convective phenomena and their hazards. The study combines radar, surface, lightning, and video data — offering a rare and valuable integrated analysis.

However, the paper suffers from issues related to language clarity, structure, figure readability, and scientific interpretation depth. The physical mechanisms are well addressed but sometimes repetitive. Strengthening the discussion, scientific context, and comparative analysis with past cases would make the paper more impactful.

**Abstract**

- 1. The abstract is too detailed and technical, with numerical values that distract the reader from the main conclusions. It reads more like part of the "Results" section. I suggest rewriting the Abstract to emphasize the research question, data fusion novelty, and main findings, while keeping numbers minimal.
- 2. The abstract lacks a clear statement of novelty what is new beyond earlier case studies such as the "Eastern Star" or previous Guizhou events. There should be a clear linkage between the event analysis and broader implications (e.g., warning systems, hazard prediction, or risk management).
- 3. Replace terms like "verifying the coupling disaster-causing mechanism of dynamic attenuation and microphysical transport" with simpler phrasing for better readability.
- 4. Improve grammar and sentence structure for flow.

**Introduction**

- 1. The introduction provides a good historical overview of downburst studies but lacks a clear research gap statement. Specify what is missing in current literature that this paper addresses (e.g., lack of integrated radar-lightning-video analysis for inland water disasters).
- 2. The link between ship disasters and meteorological mechanisms should be strengthened currently it shifts abruptly between meteorology and maritime context.
- 3. Revise for smoother flow. The introduction has long paragraphs, mixes definition, examples, and motivation.
- 4. 4-40 km downbursts are also referred to as microbursts, please correct the typo error.
- 5. First two paragraphs of Introduction include repetitive explanation of downbursts, please revise to avoid repetition.

**Results**

- **Section 4.1**: The radar analysis is comprehensive, but much of the text is descriptive rather than interpretative. It would strengthen the section to emphasize how radar features (REF, ZDR, VEL) jointly reveal the physical evolution of the micro-downburst.
- **Section 4.2**: (i) Some statements, such as "typical extremely strong wind characteristics," are qualitative. Consider including standard thresholds or reference criteria from prior downburst studies to support such claims.
- (ii) The paragraph describing the lag between wind direction reversal and temperature change could be made more analytical for instance, quantifying the time offset between dynamic and thermal responses.
- **Section 4.3**: (i) The dominance of cloud flashes (85%) is noted but not physically explained. Please discuss why cloud-to-ground lightning is suppressed in cold pool environments and how that indicates microburst maturity.
- (ii) Discuss the potential predictive role of lightning data can such lightning characteristics be used in operational downburst detection or warning?
- **Section 4.4**: (i) Clarify how the transition from northwesterly to southeasterly winds was inferred from visual cues e.g., which camera angle or scene element was used as reference.
- (ii) The timestamps from cameras (e.g., 08:18–08:40) should be explicitly cross-referenced with radar and surface observations to confirm temporal consistency.

**Figures**

- 1. Numerous figures are not referred to in the main manuscript. Please ensure that all figures are appropriately cited and discussed within the text to enhance clarity and coherence.
- 2. Please provide a detailed caption of Figure 3 which highlights the difference between 6 panels of the Figure.

**Minor Comments:**

Grammatical errors are found throughout the manuscript. Please carefully proofread the text for verb-tense consistency and sentence structure (e.g., "it has a complex formation mechanisms"  $\rightarrow$  "it has complex formation mechanisms").

Line 315: Correct spelling of "vedio" to "video."

Define all observational variables and abbreviations upon first use (e.g., REF, ZDR, VEL, CG, RCS, etc.) and maintain consistency thereafter.

Line 200: Replace "however, ....." with "However, both...." for proper sentence initiation.

Line 40–45: Reorganize sentence for clarity; replace "which may, lead to loss of course control or even capsize" with "which may lead to loss of course control or even capsizing."

Line 115–120: Replace "was lifed" with "was lifted."

Ensure consistent formatting of references — use uniform punctuation, year placement, and DOI formatting (some have inconsistent spacing).